# Efficient Residual Learning with Mixture-of-Experts for Universal Dexterous Grasping

**Ziye Huang**[2]**, Haoqi Yuan**[1]**, Yuhui Fu**[1]**, Zongqing Lu**[1,3][†]

[1]School of Computer Science, Peking University
[2]School of EECS, Peking University
[3]Beijing Academy of Artificial Intelligence

## Abstract

Universal dexterous grasping across diverse objects presents a fundamental yet formidable challenge in robot learning. Existing approaches using reinforcement learning (RL) to develop policies on extensive object datasets face critical limitations, including complex curriculum design for multi-task learning and limited generalization to unseen objects. To overcome these challenges, we introduce ResDex, a novel approach that integrates residual policy learning with a mixture-of-experts (MoE) framework. ResDex is distinguished by its use of geometry-agnostic base policies that are efficiently acquired on individual objects and capable of generalizing across a wide range of unseen objects. Our MoE framework incorporates several base policies to facilitate diverse grasping styles suitable for various objects. By learning residual actions alongside weights that combine these base policies, ResDex enables efficient multi-task RL for universal dexterous grasping. ResDex achieves state-of-the-art performance on the DexGraspNet dataset comprising 3,200 objects with an 88.8% success rate. It exhibits no generalization gap with unseen objects and demonstrates superior training efficiency, mastering all tasks within only 12 hours on a single GPU. For further details and videos, visit our project page.

## 1 Introduction

Dexterous robotic hands (Pons et al., 1999; Shaw et al., 2023) provide advanced capabilities for complex grasping tasks, similar to those performed by human hands. However, achieving universal dexterous grasping across a wide range of objects remains a significant challenge due to the high degrees of freedom (DoFs) for dexterous hands and the high variability in object geometry in the real world. Previous works (Qin et al., 2022a; Agarwal et al., 2023) develop dexterous grasping policies using reinforcement learning (RL), but these policies are limited to a small range of objects that are similar to the training objects. To improve the scalability of universal dexterous grasping, recent studies (Chao et al., 2021; Wang et al., 2023; Hang et al., 2024) introduce datasets that contain a wide variety of objects, each labeled with grasping poses. Xu et al. (2023); Wan et al. (2023); Wu et al. (2024a) leveraged these datasets to learn universal grasping policies through a teacher-student framework, which addresses the challenges of multi-task optimization. They first train state-based policies using RL to master all objects within the dataset, and then distill these policies into a universal vision-based policy.

However, these approaches exhibit certain limitations. UniDexGrasp (Xu et al., 2023) involves a complicated curriculum learning design, requiring iterative training across an expanding set of objects, which significantly increases training time and necessitates careful design for the curriculum. Similarly, UniDexGrasp++ (Wan et al., 2023) requires training various state-based policies on a large number of object clusters. This not only consumes substantial training time but may also lead to overfitting, as training is conducted individually on separate object groups. In this study, we investigate *how to directly learn a multi-task dexterous grasping policy across thousands of objects*, which enables both efficient learning and enhanced generalization.

---

[†]Correspondence to Zongqing Lu <zongqing.lu@pku.edu.cn>.

Residual policy learning (Silver et al., 2018; Johannink et al., 2019) offers an efficient approach to learning challenging tasks by training a policy to output residual actions using RL, where a suboptimal base policy is provided. This approach has the potential to address the optimization challenges in multi-task RL (Wu et al., 2024b), particularly when the base policy can effectively explore all tasks. Motivated by this, we propose **ResDex** to train a residual multi-task policy for universal dexterous grasping. The key question then becomes, *how to efficiently acquire a base policy that possesses some generalizability to grasp a wide range of objects?* Directly applying multi-task RL to all objects leads to worse results due to multi-task gradient interference (Yu et al., 2020) and requires extensive training time. Conversely, training a policy to grasp a specific object often results in poor generalization to unseen objects.

Recent work (Agarwal et al., 2023) suggests that a blind grasping policy, relying solely on robot proprioception, can robustly grasp unseen objects placed close to the palm. This is because the policy does not overfit to specific object information, leveraging feedback from the hand's joint angles and fingertip forces to adapt to various object geometries inherently. Given this insight, we propose training geometry-agnostic base policies that only observe proprioception and the 3D positions of objects to infer the object location. Experimental results demonstrate that our geometry-agnostic policy, even trained on a single object, generalizes better to a broad range of objects compared to policies with full object perception.

To enhance the diversity of grasping poses across various objects, we introduce a mixture-of-experts (MoE) approach that learns multiple base policies to represent different grasping styles. We use geometric clustering to categorize all objects and train a geometry-agnostic policy for each cluster's center. In our multi-task learning framework, we train a residual policy that not only outputs residual actions but also assigns weight to each base policy. The final control action for the robot is determined by a weighted sum of the base policies' actions and the residual action. This method effectively diversifies grasping poses by varying the weights for the base policies, thereby adapting to different object geometries.

ResDex achieves state-of-the-art training performance and generalization capabilities, successfully grasping 3,200 objects in DexGraspNet (Wang et al., 2023). It achieves a success rate of 88.8% across all training objects and exhibits **no generalization gap** when applied to unseen objects and categories. Additionally, ResDex demonstrates remarkable training efficiency, mastering such a wide range of tasks in only 12 hours on a single NVIDIA RTX 4090 GPU. In our ablation study, we highlight the critical roles of residual policy learning and geometry-agnostic experts in enhancing multi-task learning efficiency and generalization. We also demonstrate the importance of the MoE approach in achieving proper grasping poses.

Our main contributions can be summarized as follows:

- We introduce ResDex, a novel residual policy learning approach that significantly addresses the problem of efficient multi-task learning and generalization for universal dexterous grasping.
- Our technical contributions lie in the novel combination of residual multi-task reinforcement learning, geometry-agnostic base policies, and a mixture of experts framework, which together enable the development of a more generalizable and effective grasping policy.
- ResDex achieves state-of-the-art performance on the DexGraspNet dataset, demonstrating its superior training performance and generalization capabilities compared to existing methods.

## 2 RELATED WORK

**Dexterous Grasping** (Pons et al., 1999; Kappassov et al., 2015) continues to be a formidable challenge, given the high degrees of freedom in multi-fingered robotic hands and the complex geometries and physical properties of real-world objects. A fundamental task in dexterous grasping is to generate grasping poses. Recent studies have employed various methods such as contact points (Shao et al., 2020; Wu et al., 2022), affordance maps (Brahmbhatt et al., 2019; Jiang et al., 2021), natural hand annotations (Wei et al., 2023; Hang et al., 2024), and grasping datasets (Chao et al., 2021; Wang et al., 2023) to train models for synthesizing hand grasping poses. While generating target grasping poses is crucial, successfully completing a grasp also requires close-loop policies that can manage

the entire trajectory. In learning dexterous grasping policies, both imitation learning (Qin et al., 2022b; Mandikal & Grauman, 2022) and reinforcement learning (RL) (Rajeswaran et al., 2017; Wu et al., 2024b; Yuan et al., 2024; Zhou et al., 2024; Zhang et al., 2025) have shown promise. The latter offers scalable advantages across a variety of objects due to its independence from human data collection and the efficiency of simulation environments (Makoviychuk et al., 2021). Recent advancements in research explore universal dexterous grasping using RL for thousands of objects. UniDexGrasp (Xu et al., 2023) and UniDexGrasp++ (Wan et al., 2023) introduce curriculum learning and a teacher-student framework to enable training on numerous objects. UniDexFPM (Wu et al., 2024a) extends these approaches to universal functional grasping tasks. In our study, we propose an improved RL method for universal dexterous grasping that is more efficient and demonstrates superior performance and generalizability.

**Residual Policy Learning** provides an effective approach to learn challenging RL tasks when a base policy is available. In robotics, residual policy learning is extensively applied in both manipulation (Alakuijala et al., 2021; Davchev et al., 2022; Schoettler et al., 2020) and navigation tasks (Rana et al., 2020). Typically, the residual policy is constructed upon base policies that employ classical model-based control methods (Johannink et al., 2019; Silver et al., 2018). Garcia-Hernando et al. (2020) investigates residual policy learning based on human data. GraspGF (Wu et al., 2024b) explores residual policy learning on a pre-trained score-based generative model (Vincent, 2011). Zhang et al. (2023) and Jiang et al. (2024b) explore using residual policy learning to finetune RL policies. Barekatain et al. (2019) extends residual policy learning to adaptively reweight multiple expert policies. In our work, we adopt residual policy learning to tackle the challenges in universal dexterous grasping. Our method, which integrates residual RL with a mixture of geometry-agnostic experts, significantly improves multi-task learning to grasp diverse objects.

**Mixture-of-Experts (MoE)** is initially introduced by Jacobs et al. (1991); Jordan & Jacobs (1994) and typically comprises a set of expert models alongside a gating network (Shazeer et al., 2017; Fedus et al., 2022) that learns to weight the output of each expert. Recently, the MoE framework has gained substantial interest in fields such as natural language processing (Jiang et al., 2024a) and multi-modal learning (McKinzie et al., 2024). MoE has also been applied in RL policies (Doya et al., 2002; Peng et al., 2019), where each expert policy learns a distinct probability distribution that is subsequently integrated. Recent works (Cheng et al., 2023; Celik et al., 2024) use MoE to enhance multi-task learning in robotics. In our research, we use the MoE framework to improve the diversity of grasping poses in the multi-task learning of dexterous grasping policies. Each expert within our framework is a geometry-agnostic policy, trained on an individual object to develop a unique grasping style and achieve broad generalization across a variety of objects.

## 3 PRELIMINARIES

### 3.1 PROBLEM FORMULATION

We consider tabletop grasping tasks using a 5-fingered ShadowHand to grasp and lift objects initially placed on a table. The hand features 18 DoFs that control a total of 22 joints, including 4 coupled joints. Our goal is to enable grasping any object within a large object set, denoted as $\omega \in \Omega$. For each object, the task is formulated as a Partially Observable Markov Decision Process (POMDP) $M^\omega = \langle \mathcal{O}, \mathcal{S}, \mathcal{A}, \mathcal{T}, \mathcal{R}, \mathcal{U} \rangle$, representing the observation space $\mathcal{O}$, the state space $\mathcal{S}$, the action space $\mathcal{A}$, the transition dynamics $\mathcal{T}(s_{t+1}|s_t, a_t)$, the reward function $\mathcal{R}(s_t, a_t)$, and the observation emission function $\mathcal{U}(o_t|s_t)$, respectively. At each timestep $t$, the agent observes $o_t \in \mathcal{O}$ and takes an action $a_t \in \mathcal{A}$, then receives a reward $r_t = \mathcal{R}(s_t, a_t)$. The environment then transitions to the next state $s_{t+1} \sim \mathcal{T}(s_{t+1}|s_t, a_t)$. The agent's objective is to maximize the expected return across all objects $\sum_{\omega \in \Omega} \mathbb{E}\left[\sum_{t=0}^{T-1} \gamma^t r_t\right]$, where $T$ is the time limit and $\gamma$ is the discount factor.

For task learning in simulation, the observation $o \in \mathcal{O}$ includes: (1) Robot proprioception $\boldsymbol{J} \in \mathbb{R}^{123}$, including wrist position and orientation, joint angles (values of DoF positions) of the hand, fingertip states and forces on fingertip sensors; (2) Object pose, including position $\boldsymbol{b}^p \in \mathbb{R}^3$ and quaternion $\boldsymbol{b}^q \in \mathbb{R}^4$; (3) An object code $\boldsymbol{c}^\omega \in \mathbb{R}^{64}$, representing the object geometry via a pre-trained PointNet (Qi et al., 2017). In real-world settings, while precise object pose is unavailable, we opt to use the object point cloud $\boldsymbol{p} \in \mathbb{R}^{N \times 3}$, which contains $N$ points captured by cameras. The action $a \in \mathcal{A}$ consists of target joint angles of the hand and the 6D force applied at the wrist. Our aim is to learn

a vision-based policy $\pi_\theta^V (a_t | \boldsymbol{J}_t, \boldsymbol{p}_t, a_{t-1})$, parameterized by $\theta$, to maximize the expected return across all objects.

DexGraspNet (Wang et al., 2023) provides a dataset that associates each object with grasping proposals. Each grasping proposal is defined as a triplet $\boldsymbol{g} = (R, \boldsymbol{t}, \boldsymbol{q})$, representing the wrist's relative rotation $R \in \mathbb{SO}(3)$ and position $\boldsymbol{t} \in \mathbb{R}^3$ to the object and the hand's joint angles $\boldsymbol{q} \in \mathbb{R}^{22}$ for a successful grasp. Following Xu et al. (2023), these data can be integrated into the reward function to facilitate policy learning:

$$r_t = r_t^{task} + \alpha r_t^{proposal}, \tag{1}$$

$$r_t^{proposal} = -\|\boldsymbol{g} - \boldsymbol{g}_t\|, \tag{2}$$

where $r_t^{task}$ is a predefined reward for the grasping tasks. The reward term $r_t^{proposal}$ penalizes the distance to the grasping proposal, where $\alpha$ is a hyperparameter adjusting its weight and $\boldsymbol{g}_t = (R_t, \boldsymbol{t}_t, \boldsymbol{q}_t)$ represents the current relative pose of the hand to the object. The full description of the reward functions used in our framework is provided in Appendix A.2.

## 3.2 THE TEACHER-STUDENT FRAMEWORK FOR UNIVERSAL DEXTEROUS GRASPING

Directly optimizing the vision-based policy using RL faces challenges due to gradient interference (Yu et al., 2020) in multi-task RL and the high dimensionality of point cloud observations. Recent works (Xu et al., 2023; Wan et al., 2023; Wu et al., 2024a) have adopted a teacher-student framework in two stages to address these issues. First, a state-based policy $\pi_\phi^S (a_t | \boldsymbol{J}_t, \boldsymbol{b}_t^p, \boldsymbol{b}_t^q, \boldsymbol{c}^\omega, a_{t-1})$ is trained using privileged object information to master all tasks. Then, this policy is distilled into a vision-based policy using DAgger (Ross et al., 2011), an online imitation learning method.

To address the multi-task optimization challenge in learning the state-based policy, UniDexGrasp (Xu et al., 2023) proposed a curriculum learning approach. The RL training starts with a single object and, after a certain number of iterations, gradually includes more objects. This process continues until all objects are included and the policy achieves a high success rate. UniDexGrasp++ (Wan et al., 2023) introduced an improved method based on generalist-specialist learning (Jia et al., 2022). The entire object set is divided into groups through geometry-aware clustering. Numerous specialist state-based policies are then trained and subsequently distilled into a generalist state-based policy, with iterative training implemented through a curriculum. These methods require meticulous curriculum design and are time-consuming, as various policies are trained across different sets of objects. Additionally, their learned vision-based policies exhibit a significant decrease of about 7% in success rates when tested on unseen objects, indicating limited generalization capabilities.

## 4 METHOD

We propose ResDex, a framework that leverages residual policy learning combined with a mixture of experts to provide an efficient approach for universal dexterous grasping, significantly enhancing generalization capabilities. Figure 1 illustrates an overview of our framework.

### 4.1 LEARNING GEOMETRY-AGNOSTIC POLICIES

To enable efficient multi-task RL using residual policy learning, it is essential to build a base policy that can effectively explore all the involved tasks. Training a base policy directly on a single type of object often results in overfitting, which significantly decreases its generalizability to other objects. Conversely, training a policy on all objects using RL presents unique challenges, as different tasks can lead to gradient interference in the learning processes, making the training highly inefficient.

We propose to build base policies that can generalize effectively across a broad range of objects. Each base policy is trained on a single object. Multiple base policies can then be combined as a mixture-of-experts (MoE) to facilitate efficient multi-task learning.

Empirical insights from Agarwal et al. (2023) suggest that a blind grasping policy, trained solely on robot proprioception without specific object information, can better generalize to unseen objects. We hypothesize that limiting observations helps the policy avoid overfitting to specific object features. When a policy does not have complete information about object poses and geometric features, it

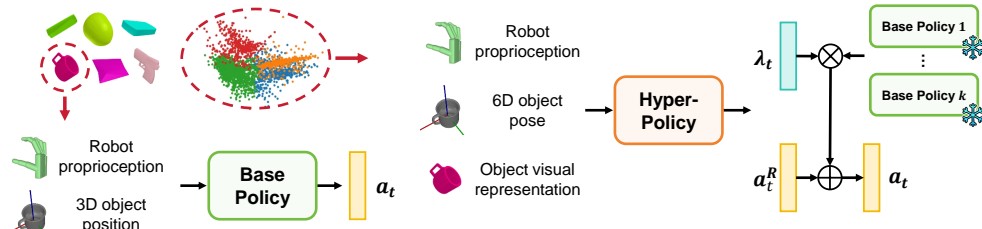

**Figure 1:** We propose ResDex, an efficient learning framework for dexterous grasping across thousands of objects. The learning process consists of two stages: (1) For each representative object from the cluster center, we train a geometry-agnostic base policy, which provides weak generalization across a broad range of objects. (2) To develop a universal policy applicable to all objects, we use residual multi-task reinforcement learning (RL) to train a hyper-policy, incorporating the base policies within a mixture-of-experts (MoE) framework. ResDex demonstrates efficient training and robust generalization to unseen objects.

tends to learn more generalizable grasping strategies and rely on the proprioceptive feedback to adjust actions. Although we cannot use a fully blind policy in our setting – as the agent must know the object's location to approach it – we integrate this insight by proposing a **geometry-agnostic base policy**, $\pi_\psi^B \left( a_t | \boldsymbol{J}_t, \boldsymbol{b}_t^p, a_{t-1} \right)$, which uses only robot proprioception $\boldsymbol{J}$ and the 3D position of the object $\boldsymbol{b}^p$.

The grasping proposal reward $r^{proposal}$ inherently leaks the object's geometric information, as the target relative wrist pose specifies "where to grasp on the object". To mitigate this unwanted information leakage and enhance generalization, we replace this term with a pose reward:

$$r_t^{pose} = -\|\boldsymbol{q} - \boldsymbol{q}_t\|, \tag{3}$$

where $\boldsymbol{q}_t$ represents the current hand joint angles. This reward encourages the hand to reach the target joint angles, focusing on the hand pose rather than the specific region to grasp on the object.

Experimental results (Section 5.3) show that our geometry-agnostic policy, trained on a single object, demonstrates remarkable generalizability to unseen objects and significantly outperforms policies that incorporate full observations or those trained using the full grasping proposal reward.

## 4.2 RESIDUAL MULTI-TASK REINFORCEMENT LEARNING

While the base policy trained on a single object offers some degree of generalizability across various objects, it typically achieves a low overall success rate. To address this, we introduce residual policy learning to develop a policy that masters all objects.

The state-based residual policy, denoted as $\pi_\phi^R \left( a_t | \boldsymbol{J}_t, \boldsymbol{b}_t^p, \boldsymbol{b}_t^q, \boldsymbol{c}^\omega, a_{t-1} \right)$, is parameterized by $\phi$. It utilizes all available state-based observations to better maximize performance in solving POMDPs.

Given the pre-trained base policy $\pi_\psi^B$, at each timestep, the base policy uses the required observations from the complete observations to compute a base action $a_t^B = \arg\max_{a_t} \pi_\psi^B \left( a_t | \boldsymbol{J}_t, \boldsymbol{b}_t^p, a_{t-1} \right)$. Simultaneously, the residual policy samples a residual action $a_t^R \sim \pi_\phi^R \left( a_t | \boldsymbol{J}_t, \boldsymbol{b}_t^p, \boldsymbol{b}_t^q, \boldsymbol{c}^\omega, a_{t-1} \right)$, and these actions are combined element-wise to form the final action $a_t = a_t^B + a_t^R$.

The generalizability of the base policy reduces the need for extensive exploration by the residual policy across diverse object geometries, making it practical to train under multi-task settings. For objects already successfully grasped by the base policy, the residual policy refines the grasping process, enhancing the success rate. For objects not successfully grasped by the base policy, the residual policy can efficiently explore in the residual action space, benefiting from the significant exploration bias provided by the base policy. We train this residual policy across the entire object set using RL, aiming to maximize the average return across all objects. Our experiments demonstrate that a single base policy, when aided by the residual policy, can achieve high success rates across thousands of objects.

## 4.3 Incorporating a Mixture of Experts

Utilizing diverse poses to grasp different objects is not only a crucial feature for dexterous hands but also essential for post-grasping manipulations in real-world tasks. While residual policy learning based on a single base policy can achieve commendable success rates, it often struggles to perform various grasping poses for different objects. This limitation arises because the base policy typically provides only a single grasping pose for its training object, thus posing a significant challenge for the residual policy to explore diverse grasping poses for certain objects.

To enhance the diversity of grasping poses, we propose a mixture-of-experts (MoE) approach. In this setup, several base policies are trained, each capable of executing distinct grasping styles, and their actions can be combined to generate a variety of novel grasping poses. To acquire base policies that exhibit diverse behaviors and grasping poses, we use geometry-aware clustering (Wan et al., 2023) to divide the object set into $k$ clusters based on object shape representations. Objects at the cluster centers are used to train the base policies $\{\pi^B_{\psi_i}\}^k_{i=1}$, leveraging their distinct and representative shapes to foster diversified grasping styles.

In multi-task learning, to integrate the base policies while learning residual actions, we replace the residual policy with a **hyper-policy**, denoted as $\pi^H_\phi\left(a^R_t, \boldsymbol{\lambda}_t | \boldsymbol{J}_t, \boldsymbol{b}^p_t, \boldsymbol{b}^q_t, \boldsymbol{c}^\omega, a_{t-1}\right)$. This hyper-policy predicts the residual action $a^R_t \in \mathcal{A}$ along with a weight $\boldsymbol{\lambda}_t \in \mathbb{R}^k$ for the MoE. At each timestep, all base policies predict base actions $\{a^B_{t,i}\}^k_{i=1}$ using partial observations, and the hyper-policy samples the weights and the residual action. The final action is computed as follows:

$$a_t = a^R_t + \frac{1}{\|\boldsymbol{\lambda}_t\|}\sum_{i=1}^k \lambda_{t,i}a^B_{t,i}, \tag{4}$$

using a normalized weighted sum of base actions in addition to the residual action. This hyper-policy aims at efficiently learning diverse, natural grasping poses by adjusting the MoE weights and enhancing multi-task performance through residual learning.

## 4.4 Method Summary

Here, we outline the complete pipeline for training ResDex, which consists of three phases. The pseudocode is provided in Appendix A.1.

**Training Base Policies:** Using the entire training set of objects, we apply K-Means clustering (Lloyd, 1982) on the PointNet (Qi et al., 2017) features of objects to generate $k$ clusters. From each cluster, we select the object closest to the center and train a geometry-agnostic base policy for each object using RL, as detailed in Section 4.1.

**Training the Hyper-Policy:** We train the hyper-policy across parallel environments that span all objects in the training set, as described in Section 4.3. During the training process, the hyper-policy is continually updated while the base policies remain fixed. To cultivate diverse and effective grasping poses while maximizing success rates, we employ a two-stage reward function:

- **First stage:** We use the reward function that includes the grasping proposal reward: $r = r^{task} + r^{proposal}$. This reward function guides the policy to follow the reference grasping poses provided by the dataset, resulting in more natural and human-like grasps.

- **Second stage:** We remove the grasping proposal term in the reward function and eliminate terms that encourage approaching the object within $r^{task}$, focusing solely on terms related to object lifting and task completion. This adjustment further enhances the policy's performance by removing constraints imposed by these reward terms. Further details on the reward functions are provided in Appendix A.2.

**Vision-based Distillation:** To learn a vision-based policy $\pi^V_\theta$ that operates without privileged object information, we adopt the teacher-student framework. The state-based hyper-policy serves as the teacher, and the vision-based policy to learn acts as the student. We use DAgger (Ross et al., 2011) to train, which involves iteratively collecting trajectories with the student policy and supervising it using the teacher policy.

**Table 1: Success rates of state-based policies.** We evaluate our method on three different random seeds. The hyper-policy is trained with four geometry-agnostic base policies. We present the success rates after each multi-task training stage.

| Method | Train(%) | Test(%) | |
| --- | --- | --- | --- |
| | | Uns. Obj. Seen Cat. | Uns. Cat. |
| UniDexGrasp | 79.4 | 74.3 | 70.8 |
| UniDexGrasp++ | 87.9 | 84.3 | 83.1 |
| ResDex (stage-1) | 90.6±0.6 | 89.7±0.8 | 90.9±0.1 |
| ResDex (stage-2) | **94.6±1.6** | **94.4±1.7** | **95.4±1.0** |

## 5 EXPERIMENTS

### 5.1 EXPERIMENT SETTINGS

We evaluate the effectiveness of our method on DexGraspNet (Wang et al., 2023), a large-scale robotic dexterous grasping dataset for thousands of everyday objects. The dataset is split into one training set and two test sets, including one that contains unseen objects in the seen categories and the other that contains unseen objects in unseen categories. The training set includes 3,200 object instances, while the test sets contain a total of 241 object instances.

To train RL policies, we set up parallel simulation environments using IsaacGym (Makoviychuk et al., 2021). For vision-based distillation, we sample 512 points on each object's mesh to provide point cloud observations. We compare our ResDex with state-of-the-art methods including UniDex-Grasp (Xu et al., 2023) and UniDexGrasp++ (Wan et al., 2023).

### 5.2 MAIN RESULTS

We compare our method with baseline methods using the hyper-policy trained with four geometry-agnostic base policies, which is our best-performing policy. The ablation study on the number of base policies used to train the hyper-policy is presented in Section 5.3.

Table 1 shows that our method outperforms UniDexGrasp++ by 6.7%, 10.1%, and 12.3% on the training and test sets respectively. Unlike previous methods, our approach shows no generalization gap, achieving consistent success rates on both the training and test sets. This consistency indicates that our method can provide a grasping policy that is more robust and generalizable.

During distillation, we use the hyper-policy trained with four base policies as the teacher to learn a vision-based policy. Performance of vision-based policies are presented in Table 2. Our vision-based policy outperforms UniDexGrasp++ by 3.4%, 8.9% and 10.5% in success rates on the three object sets respectively, demonstrating strong generalization capabilities to unseen objects.

**Table 2: Success rates of vision-based policies.**

| Methods | Train(%) | Test(%) | |
| --- | --- | --- | --- |
| | | Uns. Obj. Seen Cat. | Uns. Cat. |
| UniDexGrasp | 73.7 | 68.6 | 65.1 |
| UniDexGrasp++ | 85.4 | 79.6 | 76.7 |
| ResDex | **88.8** | **88.5** | **87.2** |

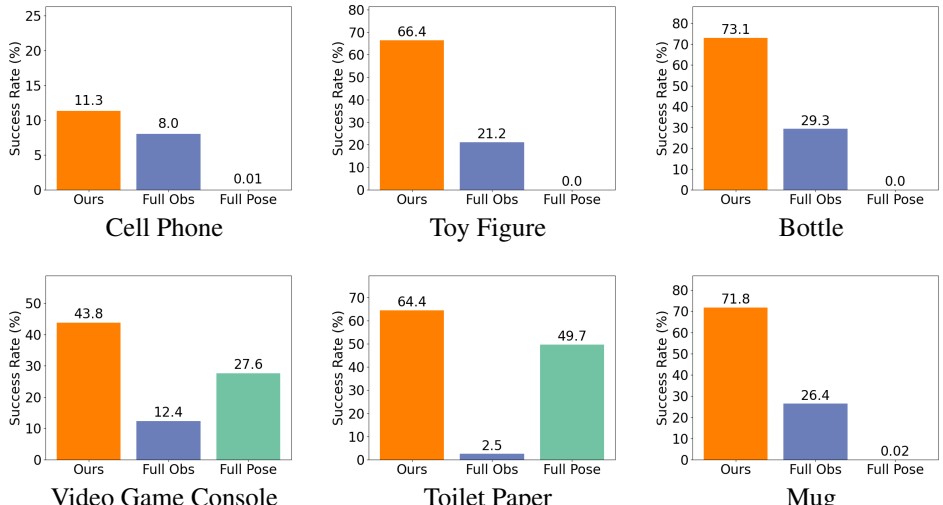

**Figure 2: Generalization performance to all objects using policies with different observations and reward, each trained on a single object.** Ours: Geometry-Agnostic policy. Full Obs: Policy trained with the complete state-based observations. Full Pose: Policy trained using the reward function that includes the full grasping proposal reward.

## 5.3 ABLATION STUDY

**Geometry-Agnostic Experts.** We compare generalizability between geometry-agnostic policies and policies trained with full state-based observations. We train 3 types of policies on 6 objects, including cell phone, toy figure, bottle, video game console, toilet paper and mug, and we evaluate their performance on the training set, which comprises more than 3000 objects. The results are shown in Figure 2. Our geometry-agnostic policies achieve higher success rates compared to other policies, achieving over 70% success rates when trained on some objects, which demonstrates remarkable generalizability. Policies with the full observations or the full grasping proposal reward demonstrate poor generalization when trained on some specific objects.

**Table 3: Ablation study on residual reinforcement learning.** We assess success rates of policies on the training set. **Method** indicates the number of base policies used. **MoE** shows the results for hyper-policies without residual actions, while **MoE+Res** shows the results for policies that output both normalized weights for MoE and residual actions.

| Method | $k = 1$ | $k = 2$ | $k = 3$ | $k = 4$ | $k = 5$ | $k = 6$ |
|---|---|---|---|---|---|---|
| **MoE** | 61.4 | 71.1 | 79.4 | 80.3 | 72.1 | 81.6 |
| **MoE+Res** | 83.2 | 82.8 | 88.1 | 90.6 | 87.6 | 88.7 |

**Residual Reinforcement Learning.** To demonstrate the multi-task learning ability provided by residual reinforcement learning, we implement an ablation method that combines base policies using a hyper-policy which only outputs the weights without residual actions. We evaluate the performance on the training set. The results, as shown in Table 3, demonstrate that for different number of base policies, the method with residual learning can notably boost the performance.

**Mixture-of-Experts.** We further demonstrate that a mixture of base policies can generate better grasping poses. We assess the quality of the grasping poses executed by our policies by computing $D = -\sum_{t=1}^{T} r_t^{proposal}$. The term $r_t^{proposal}$ is a negative reward that punishes the difference between the current grasping pose and the grasping proposal $(R, \boldsymbol{t}, \boldsymbol{q})$. Therefore, the higher the value of $D$, the less natural the grasping poses executed by the policy. The results, as shown in Table 4, reveal

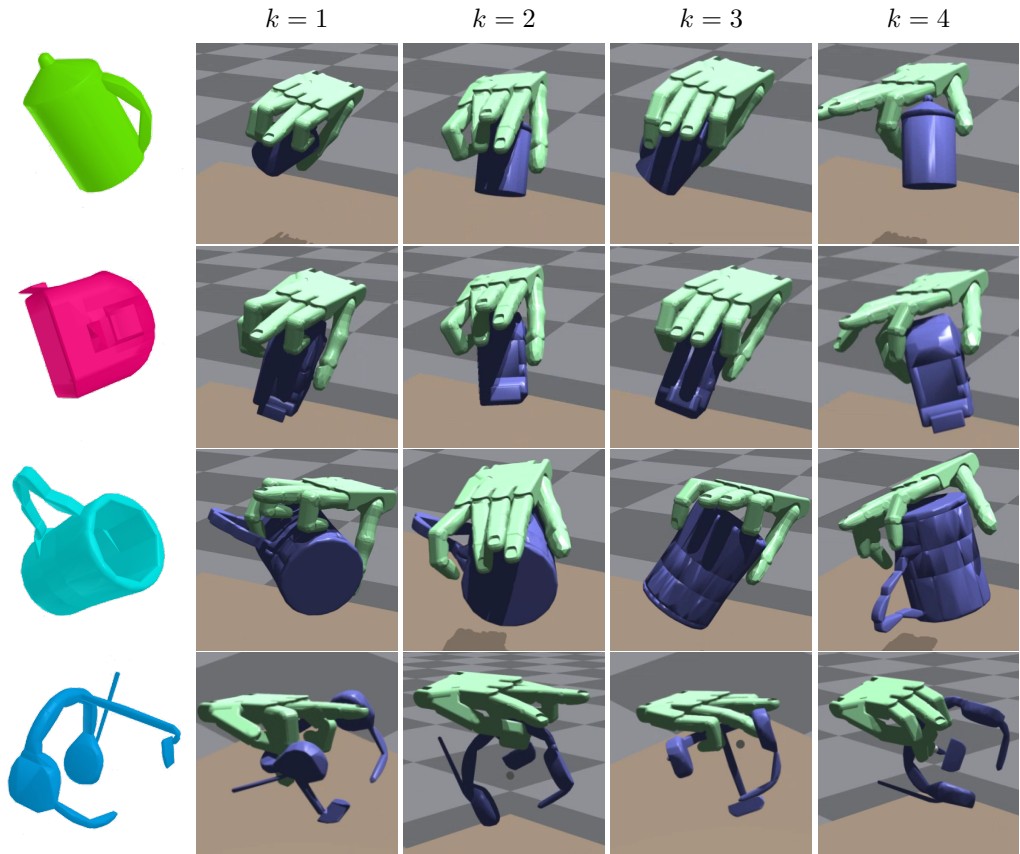

**Figure 3: Grasping poses achieved by hyper-policies trained with various numbers of base policies.** Each row displays grasping poses for a kettle, tape measure, mug, and headphone, respectively. Columns show hyper-policies trained with 1, 2, 3, and 4 base policies, arranged from left to right.

that for the hyper-policies trained over two stages, although their success rates are very close, those with more base policies generally display better grasping poses.

**Table 4: Quality of grasping poses achieved by different policies.** We evaluate the $D$ values of ResDex policies with various $k$ on the test set of unseen objects in unseen categories. The lower $D$ means the better grasping poses achieved.

| Methods | $k=1$ | $k=2$ | $k=3$ | $k=4$ | $k=5$ | $k=6$ |
|---|---|---|---|---|---|---|
| $D\downarrow$ | 223.6 | 174.5 | 194.3 | 176.3 | 204.6 | 176.1 |

Moreover, Figure 3 illustrates the various grasping poses executed by our policies with different numbers of base policies on randomly selected objects (kettle, tape measure, mug and headphones). We observe that ResDex trained with more base policies tends to learn grasping strategies that are more appropriate and natural.

**The Number of Base Policies.** We investigate how $k$, the number of base policies used to train the hyper-policy, affects the performance. Table 5 shows that ResDex with $k > 2$ consistently outperforms all the baselines according to Table 1. Furthermore, we notice that there is a gap in success rates between the configurations of $k \leq 2$ and $k > 2$. This indicates that using a mixture of base policies enables the hyper-policy to better align with the guidance provided by the grasping proposal reward. Table 6 demonstrates that the second training stage significantly boosts the success rates of our policy regardless of the value of $k$, highlighting the stability of our method.

**Table 5: Success rates of state-based policies after the first training stage.** We evaluate our method on three different random seeds. $k$ denotes the number of geometry-agnostic base policies used to train the hyper-policy.

| Method | Train(%) | Test(%) Uns. Obj. Seen Cat. | Uns. Cat. |
|---|---|---|---|
| ResDex ($k=1$) | 83.2± 1.5 | 82.8±1.0 | 85.1±0.9 |
| ResDex ($k=2$) | 82.8± 3.9 | 82.6±3.2 | 85.0±3.3 |
| ResDex ($k=3$) | 88.1± 1.2 | 88.2±0.4 | 89.3±1.0 |
| ResDex ($k=4$) | **90.6±0.6** | **89.7±0.8** | **90.9±0.1** |
| ResDex ($k=5$) | 87.6± 0.5 | 87.3±0.8 | 88.1±0.2 |
| ResDex ($k=6$) | 88.7± 0.6 | 87.8±0.5 | 88.8±1.1 |

**Table 6: Success rates of state-based policies after the second training stage.** We evaluate our method on three different random seeds. $k$ denotes the number of geometry-agnostic base policies used to train the hyper-policy.

| Method | Train(%) | Test(%) Uns. Obj. Seen Cat. | Uns. Cat. |
|---|---|---|---|
| ResDex ($k=1$) | 94.3±1.6 | 93.8±1.8 | 94.5±1.3 |
| ResDex ($k=2$) | 94.5±0.9 | 94.3±1.1 | 95.2±1.0 |
| ResDex ($k=3$) | 94.1±0.9 | 93.9±0.9 | 94.4±1.2 |
| ResDex ($k=4$) | **94.6±1.6** | **94.4±1.7** | **95.4±1.0** |
| ResDex ($k=5$) | 94.2±0.5 | 93.7±0.9 | 94.2±0.6 |
| ResDex ($k=6$) | 93.9±1.3 | 93.6±1.6 | 94.5±1.1 |

## 6 CONCLUSION AND LIMITATIONS

We propose ResDex for universal dexterous grasping, a framework that effectively addresses the challenges of training efficiency and generalization that are prevalent in existing methods. Our technical contributions include a residual policy learning framework designed for efficient multi-task reinforcement learning in dexterous grasping, a method to train geometry-agnostic base policies that enhances generalization and facilitates exploration across multiple tasks, and an MoE framework that enriches the diversity of the learned grasping poses. We demonstrate the superior performance of ResDex compared to existing methods on the large-scale object dataset DexGraspNet, notably achieving a zero generalization gap to unseen objects. The framework also showcases promising simplicity and training efficiency, marking a significant step towards scaling up dexterous learning.

The limitations of our work include: (1) Although we incorporate a grasping proposal reward to refine grasping poses, we have not yet considered the task as functional grasping. Future work could extend our approach to functional grasping tasks to further enhance general robotic manipulation in real-world settings. (2) We have not deployed the vision-based policy on hardware. Future efforts should focus on this aspect and overcome the sim-to-real gap.

The failure cases of our method arise with objects of specific sizes and shapes. For example, some large objects may unexpectedly collide with the dexterous hand upon initialization in the simulator, leading to failures. Similarly, extremely small or thin objects, such as scissors and knives, pose challenges under the tabletop grasping setting. Additionally, the policy sometimes generates unstable grasps that result in objects falling off the table before reaching the goal position. However, the policy's closed-loop nature allows itself to adapt to such cases by performing regrasping.

ACKNOWLEDGMENTS

This work was supported by NSFC under Grant 62450001 and 62476008. The authors would like to thank the anonymous reviewers for their valuable comments and advice.

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

# A    IMPLEMENTATION DETAILS

## A.1    ALGORITHM SUMMARY

---
**Algorithm 1** The Training Pipeline of ResDex

---
**Input:** Dataset $D$ of objects and grasping proposals; Number of clusters $k$.

**Preprocess:**

Apply K-means to divide objects $\omega \in D$ into $k$ clusters $C_1, C_2, \ldots, C_k$ based on PointNet features.

**for** each cluster $C_i$, $i = 1, 2, \ldots, k$ **do**

    Find the object $\omega_i$ that is closest to the center of $C_i$.

**end for**

**Training Base Policies:**

**for** $i = 1, \ldots, k$ **do**

    Train base policy $\pi^B_{\psi_i}$ on $\omega_i$, maximizing reward $r = r^{\text{pose}} + r^{\text{task}}$.

**end for**

**Multi-Task Residual RL:**

Train hyper-policy $\pi^H_\phi$ with frozen base policies $\{\pi^B_{\psi_i}\}^k_{i=1}$ on all objects.

Training stage 1: maximize $r = r^{\text{task}} + r^{\text{proposal}}$.

Training stage 2: maximize $r = r^{\text{task}} - r^{\text{reach}}$.

**Vision-based Distillation:**

Train the vision-based policy $\pi^V_\theta$ on all objects using DAgger, with frozen $\pi^H_\phi$.

---

## A.2    SIMULATION SETUP

We conduct all our experiments in IsaacGym (Makoviychuk et al., 2021), a GPU-accelerated platform for physics simulation and reinforcement learning. Each environment features a table that is 60 cm tall, with an object initialized 10 cm above the tabletop, which then falls onto it. The shadow hand is initialized 20 cm above the desktop. The task is to grasp the object and lift its center to 20 cm above the center of the tabletop.

The dataset is split into one training set and two test sets. The training set contains 3,200 object instances. The test sets include 141 instances of unseen objects within seen categories from the training set and 100 instances of unseen objects in unseen categories. For state-based policies, we use PPO (Schulman et al., 2017) for training. For vision-based policies, we distill the state-based expert policy into a vision-based policy using DAgger (Ross et al., 2011). Each geometry-agnostic policy is trained with 4,096 environments in parallel for 5,000 iterations. The hyper policy is trained with 11,000 environments in parallel for 20,000 iterations for every training stage. The vision-based policy is trained with 11,000 environments in parallel for 8000 iterations.

**Reward Function for Base Policy**    We use a modified goal-conditioned reward function to train geometry-agnostic base policies. The reward function is defined as:

$$r = r^{pose} + r^{task}$$

$X_{joint}$ denotes the joint angles. The $r^{pose}$ is defined as follows:

$$r^{pose} = -0.05 * \|\boldsymbol{q} - X_{joint}\|_1$$

$r^{task}$ is defined as follows:

$$r^{task} = r^{reach} + r^{lift} + r^{move} + r^{bonus}$$

The $r^{reach}$ encourages the hand to reach the object, as it penalizes the distance between the object and different parts of the hand. Here, $X_{obj}$ and $X_{hand}$ denote the position of the object and the hand, and $X_{finger}$ denotes positions of all the fingers. The $r^{reach}$ is defined as follows:

$$r^{reach} = -1.0 * \|X_{obj} - X_{hand}\|_2 - 0.5 * \sum \|X_{obj} - X_{finger}\|_2$$

The $r^{lift}$ encourages the hand to lift the object. It gives a positive reward when this condition can be satisfied: $f_1 = \mathbf{1}\left(\sum \|X_{obj} - X_{finger}\|_2 \le 0.6\right) + \mathbf{1}\left(\|X_{obj} - X_{hand}\|_2 \le 0.12\right)$. $a_z$ is the scaled force applied to the hand root along the z-axis. The $r^{lift}$ is defined as follows:

$$r^{lift} = \begin{cases} 0.1 + 0.1 * a_z & \text{if } f_1 = 2 \\ 0 & \text{otherwise} \end{cases}$$

The $r^{move}$ encourages the hand to move the object to the target position. $X_{target}$ denotes the target position. It gives a positive reward when this condition is satisfied: $f_2 = \mathbf{1}\left(\sum \|X_{obj} - X_{finger}\|_2 \le 0.6\right) + \mathbf{1}\left(\|X_{obj} - X_{hand}\|_2 \le 0.12\right) + \mathbf{1}\left(\|q - X_{joint}\|_1 \le 6\right)$. The $r^{move}$ is defined as follows:

$$r^{move} = \begin{cases} 0.9 - 2\|X_{obj} - X_{target}\|_2 & \text{if } f_2 = 3 \\ 0 & \text{otherwise} \end{cases}$$

The $r^{bonus}$ gives an extra reward when the object is close to the target position. We denote $\|X_{obj} - X_{target}\|_2$ as $d_{obj}$. The $r^{bonus}$ is defined as follows:

$$r^{bonus} = \begin{cases} \frac{1}{1+10*d_{obj}} & \text{if } d_{obj} \le 0.05 \\ 0 & \text{otherwise} \end{cases}$$

**Reward Function for Hyper Policy** At the first training stage for a hyper policy, we use the goal-conditioned reward function exactly the same as the one proposed in UniDexGrasp(Xu et al., 2023).

At the second training stage for a hyper policy, we use a loosened reward function defined as follows:

$$r = r^{lift} + r^{move} + r^{bonus}$$

The definitions of $r^{lift}$ and $r^{bonus}$ are the same as those mentioned above. The $r^{move}$ has loosened its condition. It is defined as follows:

$$r^{move} = \begin{cases} 0.9 - 2\|X_{obj} - X_{target}\|_2 & \text{if } f_1 = 2 \\ 0 & \text{otherwise} \end{cases}$$

### A.3 TRAINING DETAILS

**Network Architecture** We use a MLP architecture which consists of 4 layers (1024, 1024, 512, 512) for base policies and the hyper policy. For the vision-based policy, we use a simplified PointNet (Qi et al., 2017) encoder to represent the object point cloud and apply MLPs with the same hidden layer sizes for the actor and the critic. We use ELU (Clevert, 2015) as the activation function.

**Training Device and Training Time** All the state-based policies are trained on on a single NVIDIA RTX 4090 GPU. Training a base policy takes about 20 minutes, while training a hyper-policy takes about 11 hours. For the vision-based policy, we train on a single A800 GPU, taking about 16 hours.

**Analysis of Training Efficiency** To demonstrate the training efficiency of our method compared with UniDexGrasp and UniDexGrasp++, we provide a comparative analysis based on the number of training rounds, as detailed in their papers. UniDexGrasp implements a progressive training strategy — starting with a single object, expanding to several objects within the same category, and finally covering the full training set — requiring three multi-task training stages in practice. UniDexGrasp++ is more complex, involving the training of 20 multi-task policies along with several distillation stages. In contrast, our method only necessitates the training of a single multi-task policy in one trial, using between one to six low-cost, single-task base policies. Our approach is not only simpler but also efficient. As demonstrated in our experiments, our method achieves high success rates even with just one base policy. Table 7 compares the training efficiency of different methods in terms of the number of training rounds.

The hyperparameters of PPO and DAgger are described in Table 8 and Table 9.

**Table 7:** Comparison of the number of training rounds required by different methods.

| Method | Rounds for Single-Object Training ($< 20$ minutes) | Rounds for Multi-Task Training ($1 \sim 10$ hours) |
|---|---|---|
| UniDexGrasp | 0 | $\geq 3$ |
| UniDexGrasp++ | 0 | $\geq 20$ |
| ResDex | $1 \sim 6$ | 1 |

**Table 8:** Hyperparameters of PPO.

| Name | Symbol | Value |
|---|---|---|
| Episode length | -- | 200 |
| Num. envs (base policy) | -- | 4096 |
| Num. envs (hyper-policy) | -- | 11000 |
| Parallel rollout steps per iteration | -- | 8 |
| Training epochs per iteration | -- | 5 |
| Num. minibatches per epoch | -- | 4 |
| Optimizer | -- | Adam |
| Clip gradient norm | -- | 1.0 |
| Initial noise std. | -- | 0.8 |
| Clip observations | -- | 5.0 |
| Clip actions | -- | 1.0 |
| Learning rate | $\eta$ | 3e-4 |
| Discount factor | $\gamma$ | 0.96 |
| GAE lambda | $\lambda$ | 0.95 |
| Clip range | $\epsilon$ | 0.2 |

**Table 9:** Hyperparameters of DAgger.

| Name | Symbol | Value |
|---|---|---|
| Episode length | -- | 200 |
| Num. envs | -- | 11000 |
| Parallel rollout steps per iteration | -- | 1 |
| Training epochs per iteration | -- | 5 |
| Num. minibatches per epoch | -- | 4 |
| Optimizer | -- | Adam |
| Clip observations | -- | 5.0 |
| Clip actions | -- | 1.0 |
| Learning rate | $\eta$ | 3e-4 |
| Clip range | $\epsilon$ | 0.2 |

# B ADDITIONAL RESULTS

## B.1 RESULTS ON YCB DATASET

To further demonstrate the generalizability of our method, we test the learned policy on the YCB Dataset (Calli et al., 2017), which composes 75 objects. We evaluate our vision-based policy, trained on DexGraspNet, in a zero-shot manner. It achieves a success rate of 65.55%, which underscores the strong generalizability of our policy to unseen datasets. It is important to note that 30% of YCB objects are very flat and thin, which significantly challenges tabletop grasping. Additionally, because the models of YCB objects are scanned from real-world objects, they often feature irregular, non-convex shapes. This leads to differences between visual observations and collision meshes in IsaacGym, increasing the difficulty for the grasping policy, which relies on visual point clouds but interacts with mismatched physical shapes.

## B.2 RESULTS ON LEAP HAND

We additionally evaluate ResDex on the low-cost LEAP Hand, which is more accessible in laboratories. We implement a simulation setup for the LEAP Hand attached to a 6-DoF robot arm that is fixed on a table. The action space includes PD control targets for both the hand joints and the six arm joints. This setup enhances the practicability for sim-to-real deployment.

We train ResDex without modifying any hyperparameters and achieved an average success rate of 60.71% on the 3.2K objects in DexGraspNet. Several factors affect the LEAP Hand's performance, which is lower than that of the ShadowHand: (1) LEAP Hand is significantly larger and has thicker fingertips, posing challenges for grasping small objects in DexGraspNet; (2) LEAP Hand policies are trained without the grasping proposal reward due to the absence of corresponding data; (3) LEAP Hand has fewer degrees of freedom compared to ShadowHand, which can limit its capabilities; (4) The attachment to a robot arm reduces the effective workspace and alters the mechanism for controlling wrist pose, potentially affecting training performance.

Figure 4 visualizes the setup for the simulation and the learned grasping poses of the LEAP Hand.

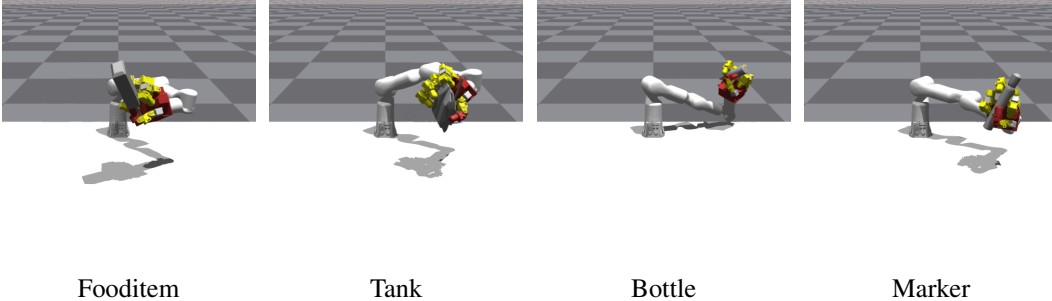

|  Fooditem | Tank | Bottle | Marker |

**Figure 4: The learned grasping behaviors of the LEAP Hand for different objects.** Using the low-cost LEAP Hand attached to a robot arm, this configuration offers greater accessibility for sim-to-real deployment.

## B.3 DIVERSITY OF THE LEARNED MoE WEIGHTS

We demonstrate that the diversity of the learned $\lambda$ correlates with the diversity of grasping styles, as evidenced by Table 4. To further investigate the diversity of the normalized weights $\lambda$ produced by the hyper-policy, we calculate the sum of each dimension, which corresponds to each base policy, throughout the evaluation process across the entire object set. The results are presented in Figure 5.

When $k = 4$ and $k = 6$, the hyper-policy can utilize different base policies. In contrast, it almost exclusively selects one base policy when $k = 5$. The result is consistent with the result in Table 4, which shows that the quality of grasping is relatively low for $k = 5$ compared to $k = 4$ and $k = 6$. Since our reward functions have no limit on the diversity of $\lambda$, there could be many reasons for these

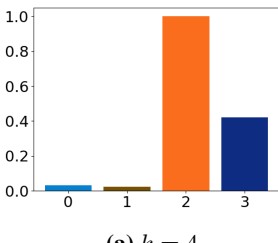 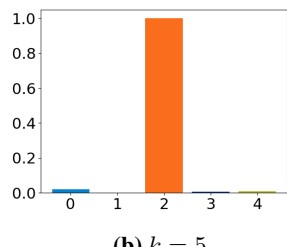 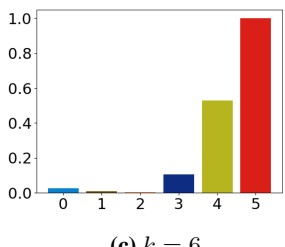

**(a)** $k = 4$          **(b)** $k = 5$          **(c)** $k = 6$

Figure 5: **Diversity of the learned** $\lambda$. $k$ denotes the number of geometry-agnostic base policies used to train the hyper-policy. Each number on the x-axis corresponds to a specific base policy. The height of each bar indicates the sum of weights for the corresponding base policy, with the largest sum normalized to 1 and other bars adjusted proportionally.

results. We suggest that explicitly diversifying $\lambda$ during the training process might lead to polices that exhibit more natural grasping styles.

Furthermore, we demonstrate the variation of $\lambda$ along the executed trajectories for different objects. As shown in Figure 6, $\lambda$ varies at different timesteps and the variation of $\lambda$ follows distinct patterns for different objects, which indicates that the hyper-policy leverages actions from different base polices to grasp different objects. Instead of collapsing to one-hot vectors or constant vectors, $\lambda$ exhibits diversity both across different objects and different timesteps within a grasping trajectory.

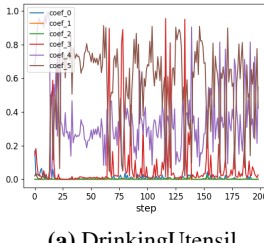 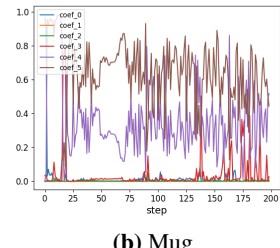 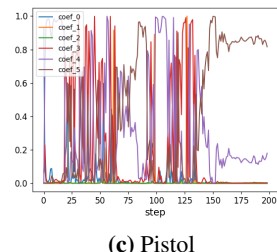

**(a)** DrinkingUtensil          **(b)** Mug          **(c)** Pistol

Figure 6: **The learned** $\lambda$ **for different objects on their executed trajectories.** The figures show the variation of $\lambda$ through grasping trajectories for different objects (drinking utensil, mug, and pistol) when $k = 6$. Weights for each base policy are plotted using a distinct color.

