# OpenReview forum: "Efficient Residual Learning with Mixture-of-Experts for Universal Dexterous Grasping"
_ICLR.cc/2025/Conference — ICLR 2025 Poster_

### Official Review · Reviewer_BKwA · 2024-10-20

**Soundness:** 3
**Presentation:** 3
**Contribution:** 3
**Rating:** 6
**Confidence:** 4

**Summary:**

In this work, the authors introduce ResDex, which integrates residual policy learning with a Mixture-of-Experts (MoE) framework for learning universal dexterous grasping policies. The method addresses drawbacks in conventional methods such as UniDexGrasp and UniDexGrasp++, including limited generalization and complex multi-task curriculum design, by leveraging geometry-unaware base policies. ResDex achieves efficient training and superior generalization, performing state-of-the-art on the DexGraspNet dataset.

**Strengths:**

1. Thorough experiments: The experiments are comprehensive, including comparisons with baselines and ablation studies that validate the importance of the method's components.
2. Performance: ResDex demonstrates state-of-the-art success rates on the DexGraspNet dataset, achieving 88.8% success in grasping unseen objects.
3. Clarity: The method is well-explained, and the presentation is enhanced by figures and tables that clearly illustrate key components of the approach.

**Weaknesses:**

1. Complexity of Approach: While simpler than UniDexGrasp and UniDexGrasp++, the combination of multiple base policies and MoE adds complexity, which goes against the original spirit of residual RL to reduce exploration burden.
2. Training Efficiency: The claim of training efficiency is not substantiated through controlled experiments. Although training times are given in the appendix, there is no comparison to baselines using comparable parameter counts and hardware.
3. Generalizability: While generalization is a key claim, the evaluation is limited to simulation on DexGraspNet data. In contrast, both UniDexGrasp and UniDexGrasp++ evaluated generalizability in different experimental settings, providing stronger support for their claims.
4. Minor Writing Issues: There are some citation issues (e.g., misuse of \citep vs. \citet in lines 101-102, line 296). Section 4.4 would benefit from a \begin{algorithm}. Additionally, the term "geometry-unaware" could be more appropriately named "geometry-agnostic."

**Questions:**

1. How is $g$ in equation 2 sampled? Will randomly sampling $g$ cause gradient interference?
2. In lines 160 and 237, are $q$ and $q_t$ hand joint positions or hand joint angle configurations?
3. One reason for using MoE is that "the base policy typically provides only a single grasping pose for its training object." Does this limitation arise due to the use of argmax for base actions (line 252)? Will other multimodal policy training methods also address this?
4. Could the authors provide more insights into how combining residual policy learning with MoE improves learning? Given that residual RL typically combines known, stable controllers with RL, what role does the MoE play? How does $\pi^H_{\phi}$ learn to weight $a^B_{t,i}$ dynamically without having $a^B_{t,i}$ as input? Could $\lambda_t$ collapse to a mean or one-hot value?

---

> ### Author Response · Authors · 2024-11-22
> **Thanks for your review! Here, we respond to your comments and address the issues. We hope to hear back from you if you have further questions!**
>
> **Q1.** The combination of multiple base policies and MoE adds complexity, which goes against the original spirit of residual RL to reduce exploration burden.
>
> **A1.** From the perspective of method simplicity, ResDex, consisting of two main components (the MoE and residual RL), is the **simplest design for universal dexterous grasping**. It avoids the complex curriculum design and proves to be insensitive to the choice of hyperparameters, as demonstrated in our experiments. Even without MoE, ResDex with a single base policy achieves a success rate of 94% (see Table 6, k=1), significantly outperforming prior methods.
>
> From the perspective of training complexity, integrating a mixture of base policies aims to increase the diversity of grasping, which **does not conflict with the spirit of residual RL**. Residual RL effectively addresses the exploration issue in multi-task optimization, improving learning efficiency whether using a single base policy (k=1) or multiple base policies within the MoE (k>1). Incorporating MoE **does not significantly increase the training cost**. As shown in Appendix A.3, adding a base policy takes only 20 minutes, which is significantly shorter than the 11 hours required to train the hyper-policy.
>
>
> **Q2.** Regarding training efficiency, there is no comparison to baselines.
>
> **A2.** We apologize for not providing the training times of baselines initially. Unfortunately, the curriculum training code for UniDexGrasp and UniDexGrasp++ has not been released, which limited our ability to perform a direct time comparison.
>
>
> However, we can provide a comparative analysis based on the number of training rounds, as detailed in their publications. UniDexGrasp implements a progressive training strategy — starting with a single object, expanding to several objects within the same category, and finally covering the full training set — requiring **three multi-task training stages**. UniDexGrasp++ is more complex, involving the training of **20 multi-task policies** along with **several distillation stages**.
>
> In contrast, our method only necessitates the training of a **single multi-task policy** in one trial, using between **one to six low-cost, single-task base policies**. Our approach is not only simpler but also efficient. As demonstrated in our experiments, our method achieves high success rates even with just one base policy.
>
> Recognizing the importance of presenting a comparison of training efficiency to baselines, we have now included this analysis in Appendix A.3.
>
>
> **Q3.** The evaluation is limited to the DexGraspNet dataset. "Both UniDexGrasp and UniDexGrasp++ evaluated generalizability in different experimental settings".
>
> **A3.** We respectfully clarify that this assessment may stem from a misunderstanding. In fact, **both UniDexGrasp and UniDexGrasp++ use only the DexGraspNet dataset for evaluation**. To the best of our knowledge, DexGraspNet remains one of the largest and most diverse datasets available for dexterous grasping tasks, encompassing over 3,200 objects with varied sizes and geometric complexities. This makes it an exceptionally suitable dataset for assessing the generalizability of grasping models.
>
> To further demonstrate generalization beyond the DexGraspNet dataset, we tested our policy on **YCB objects** in a zero-shot manner, achieving a success rate of **65.55%**. This result highlights the strong generalization capabilities of our method with unseen datasets. It is important to note that 30% of YCB objects are very flat and thin, which significantly challenges tabletop grasping. Additionally, because the models of YCB objects are scanned from real-world objects, they often feature irregular, non-convex shapes. This leads to differences between visual observations and collision meshes in IsaacGym, increasing the difficulty for the grasping policy, which relies on visual point clouds but interacts with mismatched physical shapes.
>
>
> **Q4.** About minor writing issues.
>
> **A4.** Thank you for highlighting these issues! We have corrected the citation errors you pointed out.
> Additionally, we acknowledge the benefit of including pseudocode in Section 4.4. Due to page constraints, we have added this pseudocode to Appendix A.1 and have made corresponding references in Section 4.4.
> We agree that "geometry-agnostic" is a more natural expression than "geometry-unaware". Accordingly, we have updated our terminology throughout the paper. Thank you once again for your valuable suggestions.

---

> > ### Author Response · Authors · 2024-11-22
> >
> > **Q5.** How is $g$ in equation 2 sampled? Will randomly sampling $g$ cause gradient interference?
> >
> > **A5.** Each grasping proposal $g$ is associated with an object in the dataset and is integrated into the reward to guide the policy towards effective grasping poses. For each training episode, we uniformly sample one grasping proposal per object to compute the reward function. This approach is a commonly employed method of reward shaping. Gradient interference typically arises from simultaneous training across objects, which is unrelated to the method of sampling $g$ for each object.
> >
> > **Q6.** In lines 160 and 237, are $q$ and $q_t$ hand joint positions or hand joint angle configurations?
> >
> > **A6.** In our manuscript, $q$ and $q_t$ denote the joint positions of the hand, as we have explicitly defined in Section 3.1. We did not use the term "joint angle configurations". These terms refer to the same concept in robotics.
> >
> > **Q7.** Does the limitation that "the base policy typically provides only a single grasping pose" arise due to the use of argmax for base actions?
> >
> > **A7.** In reinforcement learning, modeling continuous actions with Gaussian distributions typically results in unimodal behaviors, irrespective of whether argmax or sampling methods are used. Furthermore, since the base policy is trained to grasp a single object, it does not inherently develop novel grasping poses when exposed to other objects during the multi-task training stage. This limitation has motivated our introduction of a mixture-of-experts base policies to enhance the diversity of grasping poses.
> >
> > While some novel architectures, such as Diffusion Policies [1], are capable of achieving multi-modal behaviors, integrating these architectures with our method could be explored as a future direction but is beyond the scope of our current research.
> >
> > [1] Chi, Cheng, et al. "Diffusion policy: Visuomotor policy learning via action diffusion." The International Journal of Robotics Research (2023)
> >
> > **Q8.** Questions about combining residual policy learning with MoE, the hyper-policy learning, and the collapse issue.
> >
> >
> > **A8.** In our method, combining residual policy learning with a mixture-of-experts (MoE) is designed to alleviate the exploration burden when training on a diverse set of objects. Our findings show that even a single geometry-agnostic base policy can generalize effectively across a broad range of objects, substantiating this approach.
> >
> > The primary goal of introducing MoE is to enhance the diversity and naturalness of grasping poses. Base policies trained on objects with varying geometric features develop distinct grasping styles, crucial for handling objects of diverse shapes and sizes effectively.
> >
> > For the hyper-policy, $\pi^H_\phi$, it treats the base policies as part of the environment dynamics within the RL framework, enabling it to dynamically generate weights $\lambda_t$ that maximize returns. This process does not require direct observation of the outputs produced by the MoE base policies, akin to existing work in MoE and modular networks [2,3].
> >
> > For the question about whether $\lambda$ collapses, we provide an analysis of the learned $\lambda$ in Appendix B.3. Our results demonstrate that more than two base policies are assigned positive weights in all experimental settings, and $\lambda$ varies across different objects, indicating that the hyper-policy does not collapse.
> >
> > [2] Cai, Weilin, et al. "A survey on mixture of experts." (2024).
> > [3] Yang, Ruihan, et al. "Multi-task reinforcement learning with soft modularization." NeurIPS (2020)

---

> > > ### Comment · Reviewer_BKwA · 2024-11-23
> > > **Reply to rebuttal**
> > >
> > > Thanks for the detailed response. For Q6, although the response helps clarify, I still find the expression "hand's joint positions" confusing because "position" usually means the 3D position (i.e. in $\mathbb{R}^3$) for each of the joints instead of the actual DoF value as described.
> > >
> > > Other than that, I think most of my concerns are properly addressed. I am happy to raise my rating given that the authors integrate all the necessary changes discussed above.

---

> > > > ### Author Response · Authors · 2024-11-23
> > > >
> > > > Thank you for your positive feedback! We have updated the term "joint positions" to "joint angles" throughout the paper to enhance clarity.

---

### Official Review · Reviewer_3ip3 · 2024-10-30

**Soundness:** 3
**Presentation:** 4
**Contribution:** 4
**Rating:** 8
**Confidence:** 4

**Summary:**

This paper proposes residual learning with an MoE method for generalized grasping in simulation. The proposed method includes a set of k geometry-unaware base policies and a hyper policy that learns the weights of each base policy. It also includes a residual action based on the geometry and position of the target object and the robot's proprioception.

The proposed method avoids complex curriculum design and can be trained within 12 hours on a single 4090 GPU. Its performance peaks SOTA methods and shows no performance drop when generalized to unseen objects and categories. All claims are supported by solid experimental evidence from simulation.

**Strengths:**

- This paper proposed a novel combination of residual learning and MoE for general dexterous grasping.
- The proposed method significantly outperforms SoTA grasping methods on a large-scale simulation benchmark.
- The proposed method avoids complex curriculum design and observes almost zero performance drop when generalizing to unseen objects and categories.
- The proposed method can be trained within 12 hours on a single 4090 GPU.
- Extensive experiments in simulation support the authors’ claims.
- Overall, the paper is well-organized and written.

**Weaknesses:**

- The authors did not discuss the proposed method's limitations and failure cases. It will be interesting to see and discuss what cases still challenge the proposed method.
- There is no real robot experiment to test if the learned policy adapts well to noises and challenges in the real world.
- In line 353, the authors wrote, “Increasing k leads to a slight performance gain.” This is not true, as the proposed method performs best when k=4 and the performance drops with k larger than 4. It would be better to discuss why the model performs best when k=4.
- When reading subsections 4.1 and 4.2, I am confused about whether the base policy is trained on a single object or multiple objects. The paper contains both descriptions. This confusion is quickly resolved when I discover MoE in subsection 4.3. I suggest specifying how the base policy is used early in subsection 4.1 to avoid this confusion in the future.

**Questions:**

- In line 064, what do you mean by “base policies that only observe … 3D positions of objects to infer the object location”? What’s the difference between the 3D positions of objects and the object location?
- When training the base policy, do you train it with randomized object positions? What about orientations?
- Tables 1 and 2 suggest that the proposed method’s performance peaks with four base policies. Why is it not the case that more base policies always yield better performance?
- What is the setup for the vision-based policy? How many cameras are used? How are the cameras placed? Are there any treatments for the observed point cloud before feeding it into the policy? Is the vision-based policy evaluated in simulation or on real robots?
- Around line 421 “… and we evaluate their performance on the training set”, does the training set refer to the training set of the ablation study (i.e., the six objects), or the training set of DexGraspNet?

---

> ### Author Response · Authors · 2024-11-22
> **Thanks for your review! Here, we respond to your comments and address the issues. We hope to hear back from you if you have further questions!**
>
> **Q1.** The proposed method's limitations and failure cases.
>
> **A1.** One notable limitation of our approach is its current inability to be directly applied to functional grasping tasks. This limitation stems from the fact that our base policies are trained with restricted observations, which do not adequately capture the intricacies required for fine-grained functional grasping. Future efforts could focus on extending our method to functional grasping tasks, thereby enhancing the robot's manipulation capabilities in practical settings.
>
> Failure cases arise with objects of specific sizes and shapes. For example, some large objects may unexpectedly collide with the dexterous hand upon initialization in the simulator, leading to failure cases. Similarly, extremely small or thin objects, such as scissors and knives, pose challenges under the tabletop grasping setting. Additionally, the policy sometimes generates unstable grasps that result in objects falling off the table before reaching the goal position. However, the policy's closed-loop nature allows itself to adapt to such cases by performing regrasping.
>
> We have expanded upon these limitations and failure cases in Section 6, offering a more comprehensive discussion to guide future improvements and research.
>
>
>
> **Q2.** There is no real robot experiment.
>
> **A2.** We appreciate your comment regarding the necessity of real-world validation to demonstrate the practical applicability of our method. Currently, our research has focused on algorithmic enhancements for universal dexterous grasping within a simulated environment, aligning with the experimental setups used in prior studies such as UniDexGrasp, UniDexGrasp++, and UniDexFPM. Conducting experiments in the real world presents additional complexities, particularly the significant challenge of bridging the sim-to-real gap. We fully recognize the importance of this aspect and are committed to including real-world experiments in future work.
>
>
> **Q3.** "Increasing k leads to a slight performance gain" is not accurate.  Why is it not the case that more base policies (k>4) yield better performance?
>
> **A3.** Thank you for highlighting the inaccuracy in our description. We have revised the relevant text in Section 5.3. In terms of success rates, configurations with k>2 outperform those with k=1 and k=2. However, no significant improvement is observed with further increases in $k$.  This is because higher success rates are not solely dependent on increasing k; the success rate metric does not directly evaluate the appropriateness of grasping poses in the formulated grasping task.
>
>
>
> **Q4.** About writing issues in Section 4.1 and 4.2.
>
> **A4.** We apologize for the confusion regarding the training of the base policies. To clarify, each base policy is trained on a single object. We have updated Section 4.1 to specify this and to explain that these base policies are later used in a Mixture of Experts (MoE) approach.
>
>
> **Q5.** What is the difference between the 3D positions of objects and the object location?
>
> **A5.** "3D positions of objects" and "object location" refer to the same concept — the xyz coordinates of the object. This sentence is intended to emphasize that the base policy is provided with the object xyz position to ensure it can determine where the object is located.
>
>
> **Q6.** Are base policies trained with randomized object positions and rotations?
>
> **A6.** Yes, following the settings from UniDexGrasp and UniDexGrasp++, we randomize the rotation and z-axis of the objects. As the objects fall onto the table, this randomization also leads to randomized xy positions.
>
>
> **Q7.** About the vision-based setting.
>
> **A7.** For point cloud observations, we follow the approach in GraspGF [1]. In simulation, first, object point clouds are constructed from the objects' mesh data. At each timestep, the point clouds are transformed based on the objects' poses. During training, we apply Farthest Point Sampling to sample 512 points, which are then fed into a PointNet to extract features. The PointNet is trained simultaneously with the policy during the distillation process.
> While our experiments are conducted in simulation, the same point cloud can be acquired in the real world by using four RGBD cameras to capture the object point cloud initially, followed by object pose estimation at each timestep [1].
>
>
> [1] Wu, et al. "Learning score-based grasping primitive for human-assisting dexterous grasping." NeurIPS 2023.
>
> **Q8.** Around line 421, does the training set refer to the training set of the ablation study (i.e., the six objects), or the training set of DexGraspNet?
>
> **A8.** We are referring to the DexGraspNet training set, which includes over 3,000 objects. To avoid confusion, we have updated the paper to clarify this point. The six objects used in the ablation study are solely for case study purposes.

---

> > ### Comment · Reviewer_3ip3 · 2024-11-22
> >
> > Thank you for the detailed response. I remain positive on this paper.

---

> > > ### Author Response · Authors · 2024-11-23
> > >
> > > Thank you for your positive feedback on our work!

---

### Official Review · Reviewer_HhuW · 2024-11-02

**Soundness:** 3
**Presentation:** 4
**Contribution:** 3
**Rating:** 6
**Confidence:** 4

**Summary:**

For universal dexterous grasp execution, this work introduces a residual RL policy based on a mixture of experts for the base geometry-unaware policies.  The major motivation is to address the training inefficiency and limited generalization issues in the previous work. Technically, the authors propose to combine residual RL and the mixture of experts to tackle the gradient interference issues in training multi-task RL and the limited diversity of using a single base policy. The simulated benchmarking results and ablation study demonstrate superior generalization performance against the baseline.

**Strengths:**

- This work tackles an important problem in learning-based grasping, i.e., how to learn a performant policy for grasp execution;

- The creative combination of existing ideas to address the problems in the previous work is well-motivated and sensible;

- The idea of using geometry-unware training to enhance the generalization of the base policy is interesting and meaningful;

- The presentation is easy to follow and possesses good readability;

- Comprehensive comparison and ablation study in the experiments.

**Weaknesses:**

- The definitions of different reward functions are scattered across several sub-sections. It would be more clear for the reader if they could be grouped and discussed together.

- It would be clearer to reframe the technical contributions in the draft so that the readers can grasp the key idea more conveniently. It's because the main technical contribution is to develop a novel combination of existing techniques and demonstrate its effectiveness in learning generalizable grasping policies.

-  There is no specific comparison on this aspect. It would be nicer to also compare the training time with Unidexgrasp as the authors claim that the previous approach is inefficient, and this has been addressed by the proposed idea.

- In the experiment part, for conciseness, the ablation of different numbers of experts in Tables 1 and 2 can be taken out and put into the ablation study subsection.

**Questions:**

- The actions from different base policies are summed together based on the predicted weights of the hyper policy. I am wondering about the rotation representation used in this summation as they lie in a different space than the Euclidean one.

- Is it seemingly contradictory to first perform geometry-aware clustering and then learn a geometry-unware policy? In the end, the mixture of experts is geometry-aware. For the presentation part, it would be clearer to refine the texts for such differences. For the technical part, can they be merged into a single step in a more intelligent way?

- How is the part of grasp synthesis done?

- It seems that the number of experts doesn't represent the specific grasp styles as the model performs the best with only 4 experts, which is counter-intuitive for a dataset with more than 3k objects.

---

> ### Author Response · Authors · 2024-11-22
> **Thanks for your review! Here, we respond to your comments and address the issues. We hope to hear back from you if you have further questions!**
>
> **Q1.** The definitions of different reward functions are scattered across several sub-sections.
>
> **A1.** Thank you for your feedback regarding the presentation of reward functions. We have consolidated the full details of the reward functions in Appendix A.2 and have added a directive in Section 3.1 to guide readers to these details for clarity. In the main body of the paper, we simplify our discussion on rewards to three primary notations, $r^{task}, r^{proposal}$, and $r^{pose}$, to minimize confusion.
>
> We have chosen not to present the full details of each reward term directly in the main text for several reasons:
>  - While the design of the reward functions is important, it is not the primary contribution of this work. The reward terms we use are adopted from established methodologies in prior research, particularly UniDexGrasp and UniDexGrasp++.
>  - Due to page constraints, we prioritized brevity in the main text. Detailed implementations of the reward functions are thus included in the appendix. In Section 3.1, we clarify the distinctions between $r^{task}$ , a hand-designed reward function for grasping tasks, and $r^{proposal}$, which is derived from the grasping proposals within the dataset. The term $r^{pose}$, which is a sub-component of $r^{proposal}$, is introduced in Section 4.1 based on our observations during the training of base policies.
>
>
> **Q2.** Reframe the technical contributions.
>
> **A2.** Thank you for your suggestion! We have revised the summary of our technical contributions at the end of the Introduction. Our technical contributions are grounded in the novel integration of residual multi-task reinforcement learning, geometry-agnostic base policies, and a mixture-of-experts framework, which together enable the development of a more generalizable and effective grasping policy.
>
>
> **Q3.** Comparison of training efficiency to UniDexGrasp.
>
> **A3.** We apologize for not providing the training times of baselines initially. Unfortunately, the curriculum training code for UniDexGrasp and UniDexGrasp++ has not been released, which limited our ability to perform a direct time comparison.
>
> However, we can provide a comparative analysis based on the number of training rounds, as detailed in their publications. UniDexGrasp implements a progressive training strategy — starting with a single object, expanding to several objects within the same category, and finally covering the full training set — requiring **three multi-task training stages**. UniDexGrasp++ is more complex, involving the training of **20 multi-task policies** along with **several distillation stages**.
>
> In contrast, our method only necessitates the training of a **single multi-task policy** in one trial, using between **one to six low-cost, single-task base policies**. Our approach is not only simpler but also efficient. As demonstrated in our experiments, our method achieves high success rates even with just one base policy.
>
> Recognizing the importance of presenting a comparison of training efficiency to baselines, we have now included this analysis in Appendix A.3.
>
>
> **Q4.** About presentation of the ablation of different numbers of experts.
>
> **A4.** Thank you for your suggestion! We have revised the presentation to report only the results for k=4 in the main results (Table 1). To improve the clarity of our paper, results for different values of k have been moved to the ablation section (Tables 5 and 6).
>
> **Q5.** How to handle rotation representations in the weighted summation of base policies' actions?
>
> **A5.** We use 6D force to control wrist translation and rotation. While it is true that Euler angles do not form a Euclidean space and linear interpolation between Euler angles does not typically result in a linear rotation, the weighted sum and residual actions produced by the hyper-policy are nevertheless capable of generating any required 3D torques. Since the hyper-policy dynamically assigns weights and residual actions, there is no need to explicitly define a rotation action within a Euclidean space for the purposes of our method. Regarding the finger actions, they involve individual joint positions which can be linearly interpolated.

---

> > ### Author Response · Authors · 2024-11-22
> >
> > **Q6.** The confusion about geometry-aware clustering and geometry-unaware policies. Is it possible to merge them into a single step?
> >
> >
> > **A6.** In our framework, the geometry-unaware policy and geometric clustering serve **distinct purposes** and are presented in different sections. The geometry-unaware policy, discussed in Section 4.1, aims to minimize overfitting on object geometry, thereby enhancing generalization to unseen objects. This "unawareness" implies that the base policy does not directly observe object geometry nor is it influenced by geometry-specific rewards during its training.
> >
> > Conversely, geometric clustering, introduced in Section 4.3, is utilized to develop a mixture of base policies capable of producing varied grasping styles. This strategy leverages geometric similarities to ensure that each base policy specializes in handling a specific group of object geometries, enriching the diversity of the grasping poses for multi-task learning.
> >
> > We acknowledge the potential to integrate these stages into a more unified approach, perhaps by dynamically generating object clusters as base policies are trained, which could streamline the learning process. While the current two-stage design is simple and well-motivated for the scope of this paper, your suggestion provides a promising direction for future research.
> >
> >
> > **Q7.** How is the part of grasp synthesis done?
> >
> > **A7.**  The grasping proposals are accompanied with the DexGraspNet dataset, provided by UniDexGrasp, as discussed in Section 3.1. Their synthesis process involves using a point-cloud-conditioned generative model and ContactNet. Please refer to UniDexGrasp [1] for further details.
> >
> > [1] Xu, et al. "Unidexgrasp: Universal robotic dexterous grasping via learning diverse proposal generation and goal-conditioned policy." CVPR 2023.
> >
> >
> > **Q8.** The model performs the best with only 4 experts, which is counter-intuitive for a dataset with more than 3k objects.
> >
> > **A8.** We appreciate your concern that achieving optimal performance with only 4 experts might seem counterintuitive given the dataset's size. We would like to provide the following clarifications:
> > - The effectiveness of the hyper-policy primarily stems from its capacity for residual learning, rather than solely relying on weighting actions from base policies. Leveraging our geometry-unaware base policies, which exhibit strong generalization capabilities, the hyper-policy can efficiently learn residual actions for a multitude of tasks (3,200 objects).
> > - It is important to note that the diversity of grasping styles does not directly correlate with grasp success rates. Even with a single base policy providing relatively unimodal grasping poses, our model maintains high success rates (refer to Tables 5 and 6). While increasing the number of base policies enhances the model's capacity and aids the hyper-policy in learning diverse grasping poses for different objects, this expansion does not necessarily translate to improved success rates.

---

> > > ### Comment · Reviewer_HhuW · 2024-11-26
> > > **Feedback from Reviewer HhuW**
> > >
> > > Thank you for your efforts in addressing my concern. The rebuttal has addressed my concern, and I remain in favor of accepting this paper.

---

> > > > ### Author Response · Authors · 2024-11-27
> > > >
> > > > We sincerely appreciate your review and your positive feedback on our work!

---

### Official Review · Reviewer_8Don · 2024-11-03

**Soundness:** 4
**Presentation:** 4
**Contribution:** 3
**Rating:** 8
**Confidence:** 5

**Summary:**

This work combines concepts from residual policy learning, mixture of experts and student-teacher distillation to train generalizable grasping policies with a dexterous hand in simulation. The proposed method ResDex has multiple stages of reinforcement learning in simulation, including the training of proprioception only policies on different types of objects, training of a residual mixture of experts policy and training of policies with a curriculum of reward functions. The resulting policies are shown to achieve a high performance for grasping unseen object instances and categories.

**Strengths:**

1. The authors propose a residual mixture-of-experts policy for dexterous grasping, where the individual base policies are trained on different datasets. This is both a novel and a very interesting idea. In particular, the individual policies are trained on clusters of object geometries using only proprioceptive information, whereas the high level mixing policy is trained with state information.
2. The work further includes a curriculum of two reward functions: the first reward function encourages similarity to demonstrated grasps, whereas the second reward only encourages grasping success. This is a good trade-off between encouraging natural and optimal grasps.
3. The method is compared to prior work on the reinforcement learning of dexterous grasping and it is shown to achieve a higher zero-shot grasping success rate. Appropriate ablations for the various parts of the mixture policy are included.

**Weaknesses:**

1. The paper lacks any real-world experiments. Therefore, it is not clear if the specific design decisions made in this work, which increase the performance in the simulator, lead to a higher real-world grasping success rate. Further, real-world evaluation might be challenging because the Shadow Hand is very expensive. The work could be strengthened by also running experiments with the LEAP Hand, for example, which is more accessible.

**Questions:**

Is a sophisticated robot hand necessary to reach a high performance, or could similar performance be reached with simpler hands like the LEAP or Allegro?

---

> ### Author Response · Authors · 2024-11-22
> **Thanks for your review! Here, we respond to your comments and address the issues. We hope to hear back from you if you have further questions!**
>
> **Q1.** The paper lacks real-world experiments.
>
> **A1.** We appreciate your comment regarding the necessity of real-world validation to demonstrate the practical applicability of our method. Currently, our research has focused on algorithmic enhancements for universal dexterous grasping within a simulated environment, aligning with the experimental setups used in prior studies such as UniDexGrasp, UniDexGrasp++, and UniDexFPM. Conducting experiments in the real world presents additional complexities, particularly the significant challenge of bridging the sim-to-real gap. We fully recognize the importance of this aspect and are committed to including real-world experiments in future work.
>
> **Q2.** The work could be strengthened by running experiments on LEAP Hand, which is more accessible.
>
> **A2.**  Thank you for your valuable suggestion. In response, we have implemented a simulation setup for the LEAP Hand attached to a 6-DoF robot arm that is fixed on a table. The action space includes PD control targets for both the hand joints and the six arm joints. This setup enhances the practicability for sim-to-real deployment.
>
> We trained ResDex using this setup without modifying any hyperparameters and achieved an average success rate of 60.71% on the 3.2K objects in DexGraspNet.
>
> Several factors affect the LEAP Hand's performance, which is lower than that of the ShadowHand: (1) LEAP Hand is significantly larger and has thicker fingertips, posing challenges for grasping small objects in DexGraspNet; (2) LEAP Hand policies are trained without the grasping proposal reward due to the absence of corresponding data; (3) LEAP Hand has fewer degrees of freedom compared to ShadowHand, which can limit its capabilities; (4) The attachment to a robot arm reduces the effective workspace and alters the mechanism for controlling wrist pose, potentially affecting training performance.
>
> We present the detailed results for LEAP Hand with the robot arm setup in Appendix B.2.

---

> > ### Comment · Reviewer_8Don · 2024-11-25
> > **Response**
> >
> > Thank you for conducting an additional experiment! I remain in favor of accepting this paper.

---

> > > ### Author Response · Authors · 2024-11-26
> > >
> > > Thank you! We sincerely appreciate your positive feedback.

---

### Meta-Review · Area_Chair_Hisd · 2024-12-21

**Metareview:**

ResDex is a framework for dexterous grasping that combines residual policy learning and a mixture-of-experts (MoE) approach. Each base policy is trained on clusters of objects using only proprioceptive data, while a hyper policy fuses these base policies with a small residual adjustment. This design circumvents complex multi-task curricula, showing 88.8% success on a 3,200-object dataset (DexGraspNet), no performance gap on unseen objects, and efficient training (12 hours on a single GPU).

Strengths:
-- The method effectively tackles multi-object grasping by mitigating gradient interference and leveraging geometry-unaware base policies.

-- Demonstrates robust performance on unseen objects without additional fine-tuning.

-- Achieves state-of-the-art results within 12 hours, surpassing prior curriculum-heavy approaches.

Weaknesses:

-- The method’s transferability to physical robots remains unverified.

-- Insufficient discussion of cases where performance might degrade.

-- The multi-stage approach (MoE + residuals) still adds overhead despite aiming to streamline training.

After carefully reading the paper, the reviews and rebuttal discussions, the AC agrees with the reviewers on recommending to accept the paper.

**Additional Comments On Reviewer Discussion:**

The weaknesses are described above. The authors have addressed most comments in rebuttal and the reviewers generally agree to accept the paper.

---

### Decision · Program_Chairs · 2025-01-22

Accept (Poster)